# *Diaporthe* and *Diplodia* Species Associated with Walnut (*Juglans regia* L.) in Hungarian Orchards

Andrea Zabiák [1,2], Csilla Kovács [3], Ferenc Takács [3], Károly Pál [1], Ferenc Peles [1], Erzsébet Fekete [4], Levente Karaffa [4,5], Kata Mihály [1], Michel Flipphi [4] and Erzsébet Sándor [1,*]

1    Institute of Food Science, Faculty of Agricultural and Food Science and Environmental Management, University of Debrecen, H-4032 Debrecen, Hungary
2    Kálmán Kerpely Doctoral School, University of Debrecen, H-4032 Debrecen, Hungary
3    Research Institute Újfehértó, Agricultural Research and Educational Farm, University of Debrecen, H-4244 Újfehértó, Hungary
4    Department of Biochemical Engineering, Faculty of Science and Technology, University of Debrecen, H-4032 Debrecen, Hungary
5    Institute of Metagenomics, University of Debrecen, H-4032 Debrecen, Hungary
*    Correspondence: karaffa@agr.unideb.hu

**Abstract:** Walnut (*Juglans regia* L.) production is a developing sector in Hungarian horticulture, where preharvest fruit rot increasingly causes economic losses. Previously, fungi from the Diaporthaceae and Botryosphaeriaceae families were isolated with a high incidence from rotten fruits. Here, we identify these pathogens from different plant parts (twigs, buds, and shoots) in orchards in the north-east of Hungary, and test their pathogenicity on immature nuts. *Diaporthe eres* Nitschke and *Diplodia seriata* De Notaris were identified in isolates from different symptomatic and asymptomatic plant parts based on their ITS (nuclear ribosomal internal transcribed spacer) and *tef1* (translation elongation factor EF-1-*alpha* gene) DNA sequences. Purified monocultures caused rot of immature nuts following in vitro inoculation. Our results suggest that *D. eres* Nitschke and *D. seriata* De Notaris lingering on buds and overwintering woody parts may affect the seasonal nuts through wound infection. Infection by *Diaporthe* and *Botryosphaeriaceae* species present on woody plant parts, as well as on/in buds has been reported in Mediterranean countries. This is the first report of such stepwise aetiology from a region with continental weather. Climate change, associated weather patterns and the limitations of fungicide use in the European Union, among other factors, could be responsible for the increasing number of infections and economic damage caused by these pathogens.

**Keywords:** walnut rot; wood cancer; bud infection; *Diplodia* sp.; *Diaporthe* sp.; pathogenicity

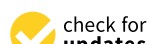



## 1. Introduction

Walnut contains considerable amounts of nutrients that contribute to a healthy diet, such as proteins and fatty acids [1]. The most common species are Black walnut (*Juglans nigra* L.) and English walnut (*Juglans regia* L.), but only the latter is cultivated for commercial purposes, because Black walnut has a hard shell and it is difficult to hull [2]. English walnut is an ancient horticultural species [3], which originates from Central Asia [4]. In Europe, the non-native tree has been cultivated to produce this fruit since 1000 BC [5]. Walnut consumption reportedly decreases the chance of developing heart diseases correlated to its poly-unsaturated fatty acid content [6–9]. Walnut kernel is rich in tocopherols (vitamin E) and essential amino acids [1,10].

Sustainable nut production depends on weather conditions, insects, and microbes. Spring frost can cause problems during the fertilization period due to the cold sensitivity of the generative organs. The damage results in paper-nutshell with hollows or even an absence of the kernel [11,12]. In addition, *Juglans regia* (L.) is frequently infected by various microorganisms that cause serious yield losses [13]. Bacterial walnut blight, caused by

*Xanthomonas* species, which is common in plantations, results in brown patches on the kernel and the husk [14,15]. Walnut anthracnose is caused by *Ophiognomonia leptostyla* (Fries) Sogonov, a fungus of Diaporthales (Sordariomycetes). It is a widespread disease. Specific symptoms of the walnut anthracnose can be detected on different plant parts (leaves, twigs, shoots, and fruit husk), and are characterized by irregular brown patches with chlorotic halos that become larger and greyish, and the husk ends up being rotten [16,17]. The disease affects the quality of the fruit and may result in premature fruit drop [12]. Brown apical necrosis causes brown colour lesions at the top of (immature) green nuts. *Xanthomonas arboricola* pv. *juglandis* (Pierce) Vauterin, Hoste, Kersters, and Swings, as well as fungi of the *Fusarium* and *Alternaria* genera (belonging to the Sordariomycetes and Dothideomycetes classes, respectively) are associated with these latter symptoms [14,18]. In recent years, nut crops have been significantly affected by fungal diseases, e.g., twig canker, dieback, shoot blight, and nut rot [19–23]. Species of the *Botryosphaeriaceae* and *Diaporthaceae* families are often reported as causative agents of such pathogenies in many woody plants, including walnut trees in several countries [21–30]. These fungal pathogens are able to survive on seasonally inactive and dead plant parts and produce asexual conidia in pycnidia. The vegetative spores infect wounds and germ on/in different plant parts, including (immature) nuts [20,29,30].

Walnut production is economically important in Hungary; its production area is currently the third largest after apple and sour cherries [31], with a continuously increasing output [32]. *Diaporthe* and Botryosphaeriaceae species have become widely disseminated in Hungarian horticulture, in commercial walnut orchards and nurseries [33,34], and in more traditional vineyards, as the cause of grapevine trunk diseases [35,36]. The prevalence of these pathogens may further be exacerbated by the acceleration of climate change [37,38], which suggests that *Diaporthe* and Botryosphaeriaceae species can become a menace to English walnut commerce in Hungary [34].

The morphological characteristics are less informative for the species-level taxonomical identification of *Diaporthe* [39] and Botryosphaeriaceae species [40]. Identification by means of comparative sequence analysis of the nuclear ribosomal internal transcribed spacer (ITS) combined with that of the large intron in the *tef1* gene (viz. the ubiquitous gene encoding translation elongation factor EF-1 *alpha*) provides easier, standardized means to assess fungal taxonomy [29,37,39–43]. The aims of the current study were (i) to detect the presence of fungal pathogens on different plant parts other than fruits (twigs, buds, and shoots) in walnut orchards in Hungary; (ii) to identify those fungi using the ITS and *tef1* molecular markers; and (iii) to test their pathogenic capacity on green, immature walnut fruit to examine potential infection routes.

## 2. Materials and Methods

### 2.1. Collection and Purification of Fungal Isolates

Samples were collected from 40 English walnut trees at four different locations, on the territory of the municipalities of Hajdúdorog, Jánkmajtis, Tarpa, and Újfehértó, during the spring of 2018 (Figure 1). Both asymptomatic and symptomatic twigs were collected from each tree. Moreover, asymptomatic and symptomatic buds or green fruits were also collected from some of the sampled trees from each of the orchards to detect possible latent infections and their sources (Table 1).

The tissue surface of plant samples was disinfected with a 10% chlorogen-sesquihydrate (Neomagnol; Parma Produkt Ltd., Budapest, Hungary) and 0.1% Tween20 (Merck KGaA, Darmstadt, Germany) solution for one minute, then the samples were washed twice in sterile distilled water [44]. Bark tissues were removed before disinfection. The treated samples were placed on malt-extract agar (MEA) solid medium (Biolab, Budapest, Hungary) containing streptomycin sulphate ($0.1 \text{ g L}^{-1}$) (Merck KGaA, Darmstadt, Germany) to minimize the growth of the resident bacteria [44]. The entire bud was cut into two and was put on the plates. In the case of woody tissues, small pieces were put on plates, as previously described in [44]. After incubation of the plates at room temperature, mycelia from the

appearing fungal colonies were transferred to fresh potato dextrose agar (PDA) medium (Biolab, Budapest, Hungary), following seven days of incubation at room temperature. The fungal cultures were subjected to several rounds of single-cell purification and were maintained as monocultures. Conidiospores were kept suspended in a 33% glycerol solution (end concentration) at −80 °C. Pure fungal cultures were used for further studies. The colour and texture of the colonies were studied as macromorphological identifiers on the PDA medium. The asexual fruiting body and the shape and size of the conidia were examined using a Zeiss AxioImager phase-contrast microscope, equipped with a AxioCam MRc5 camera (Zeiss, Oberkochen, Germany).

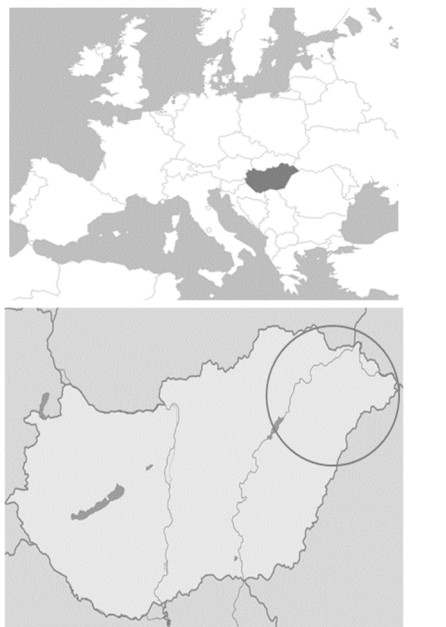
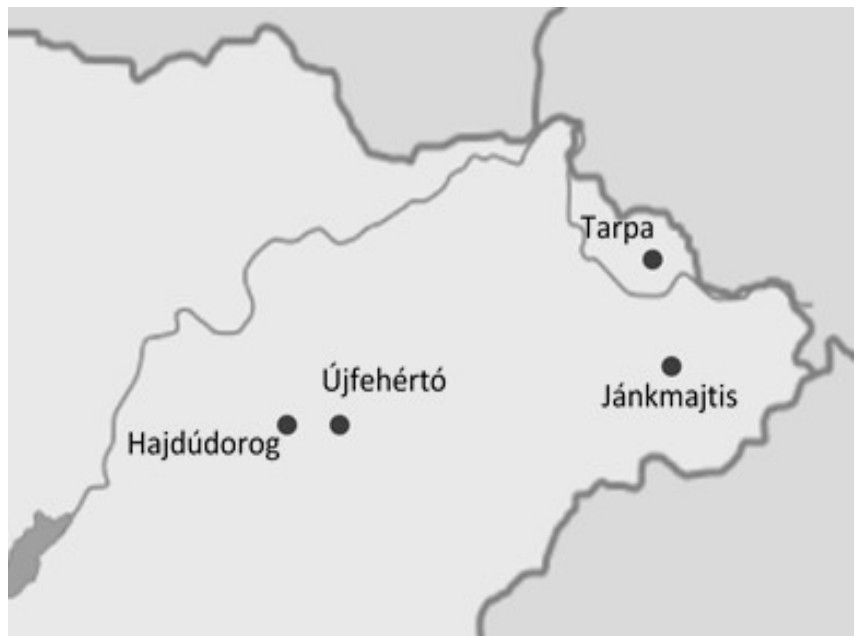

**Figure 1.** Location of the sampling sites of walnut trees in the north-east of Hungary (Szabolcs–Szatmár–Bereg and Hajdú–Bihar Counties).

**Table 1.** The occurrence of *Diplodia* sp. and *Diaporthe* sp. in the collected plant samples. Preliminary taxonomical identification was based on the morphological attributes of the colonies growing on potato dextrose agar (PDA).

| Samples | No. of Sampled Plant/Plant Part | Recovery Percentage of Pathogens from Samples [1] | |
|---|---|---|---|
| | | *Diaporthe* sp. | *Diplodia* sp. |
| Walnut tree | 40 | 17.5% | 17.5% |
| Twig | 40 | 17.5% | 12.5% |
| Bud | 9 | 0% | 11% |
| Green fruit | 7 | 0% | 14% |

[1] Ratio of samples containing the pathogen.

### 2.2. Molecular Identification

The isolates were primarily separated by the much typical morphological characteristics. However, just to be on the safe side, five of the seven isolates displaying the typical morphological characteristics of five of the seven isolates with the typical morphological characteristics of *Diaporthe* sp. (Diaporthaceae, Diaporthales, and Sordariomycetes) [27,45], and three isolates showing growth characteristics typical of members of the Botryosphaeriaceae (Botryosphaeriales and Dothideomycetes) [21] were chosen for further, sequence-based identification from the 113 fungal colonies obtained. These eight isolates came from

different orchards (Figure 1) and from different plant parts. Molecular taxonomic identification with the sequence of the ITS rDNA region allowed for determining the genus of the individual fungal monoculture [42]. Species identification was pursued analysing the sequence of the large intron in the filamentous fungal *tef1* gene encoding translation elongation factor EF-1 *alpha* (*tef1*) [39]. These marker sequences were compared with the extant data of the characterized species of the genera in question.

DNA extraction was carried out with the NucleoSpin Plant II Kit (Macherey-Nagel, Düren, Germany) according to the manufacturer's instructions. Mycelia from a seven day-old fungal colony were scraped from the surface of PDA plates and transferred to 2 mL ZR BashingBead Lysis Tubes (Zymo Research Corp, Irvine, CA, UCA) that contained 0.7 mL of 2 mm bashing beads and 500 μL Lysis Buffer 1, and subsequently thoroughly vortexed to disrupt the biomass and release fungal DNA. For species identification, DNA sequence data of the ITS locus (ITS1 and ITS4 primers) [46] and the *tef1* gene (EF-978F and EF1-986R primers) [47] were generated by targeted PCR. The PCR reaction volume was 25 μL and contained 12.5 μL DreamTaq Green Master Mix (Thermo Fisher Scientific, Germany), 0.5 μL of each locus-specific primer (10 pmol μL$^{-1}$), 10.5 μL nuclease-free water, and 1 μL of DNA solution (10 ng μL$^{-1}$). The PCR amplification started with pre-incubation for 3 min at 95 °C to melt out the double-stranded template DNA, followed by 35 cycles of 30 s at 95 °C to obtain a single-stranded template, 1 min of primer annealing at 56 °C for ITS and 55 °C for *tef1* and 1 min of primer extension at 72 °C. The program ended with one post-cyclic incubation for 15 min at 72 °C to complete the unfinished primer extension intermediates [36,43].

The PCR products were resolved in 1% agarose gel (Bioline, Memphis, TN, USA) native agarose gels containing GelRed stain (Biotium, Fremont, CA, USA) in 1 μL/mL concentration to visualize double-stranded DNA. Electrophoresis was performed for 60 min at 100 V. The amplified DNA of the expected size was cut out from the gel and subsequently purified with NucleoSpin Gel and PCR Clean-up Kit (Macherey-Nagel GmbH and Co., KG, Düren, Germany). Purified DNA was then cloned in the pGEM-T Easy vector (Promega Corporation, Madison, WI, USA). Plasmid DNA was isolated using the NucleoSpin Plasmid EasyPure kit (Macherey–Nagel). Three independent clones were chosen and sequenced over both strands using universal primers hybridizing to the vector (Microsynth Gmbh, Austria). The ITS and *tef1* sequences of our isolates were deposited at GenBank (accession numbers MN726698–726705, ON381300–381305, and OP207880–OP207879). Subsequently, the ITS and *tef1* sequences from the selected monocultures were diagnosed by comparison with ortholog sequences extant in the NCBI nt/nr and Whole Genome Shotgun contig databases and mined upon BlastN analysis [48]. Ex-type and other publicly accessible ortholog sequences were taken from the paper of Guarnaccia and co-workers [39] for the assessment of the *Diaporthe eres* species complex by means of phylogenic analysis of the ITS and *tef1* markers, and from the work of Zhang and co-workers [49] for the assessment of the *Diplodia seriata* species complex. (Supplementary Table S1). For the phylogenetic analysis, selected sequences were first aligned with Clustal-X [50]. The resulting alignment was manually checked for ambiguities and, where necessary, adjusted using Genedoc [51]. Phylogenetic analysis was performed with the MEGA 7.0 program using the curated DNA alignments as the input [52]. Maximum likelihood was used to infer trees, based on the Tamura 3-parametric matrix. Nearest-neighbor interchange (NNI) was applied as a heuristic approximation. Support for internal branches was assessed with 1000 bootstrapped iterative replications.

### 2.3. Pathogenicity Test

Pathogenicity tests were conducted to examine the ability of *Diplodia* and *Diaporthe* isolates to infect immature walnut husks and their pathogenic capacity. Pathogens were isolated from twigs, buds, and green fruits and subjected to several rounds of single-cell purification, as described above. Five *Diaporthe* (J1004, T1010, U1001, U1003, and U1008)

and three *Diplodia* (D1012, U1012, and U1013) purified isolates, cultured from the collected samples, were selected for the inoculation of prior disinfected green fruits.

Green walnuts from the "Milotai 10" cultivar without visible symptoms of necrosis were collected from the Újfehértó site. Three biological replicates as well as three controls were tested in parallel. First, the walnuts were submerged for 1 min in chlorogensesquihydrate (Neomagnol)–Tween20 solution to sterilize the surfaces. The fruits were inoculated with mycelial plugs from the margin of seven day-old fungal colonies onto a freshly inflicted lesion of the husk using a sterile cork borer [29]. In the negative control experiment, the wounded husk was mock-inoculated with a sterile PDA medium plug. Inoculated green walnuts were individually wrapped in Parafilm (Merck, Germany) to avoid dehydration of the lesion and the agar plug, and were incubated at 25 °C for three weeks [21]. Mock-inoculated fruits were incubated under the same conditions as pathogen-treated green walnuts. Subsequently, both the husks and the kernels were analysed and classified according to the severity of symptoms as a measure of the progress of necrosis. The husks were rated from "0" to "5" as follows: class 0—healthy fruit without visible symptoms of rot; 1—husk with small brownish lesion around the wound; 2—husk with more extensive brownish lesion around the inoculation zone; 3—half rotten green walnuts; 4—further progressed rot but unaffected tissue remains; 5—entirely rotten, blackened walnuts (Figure 2a–e). Kernels were classified on a scale from "0" to "3" as follows: class 0—asymptomatic green walnuts resembling the mock controls; 1—nuts discoloured; 2—walnuts with small brown spots and mycelia on the kernel; 3—entirely rotten kernels (Figure 3a–d).

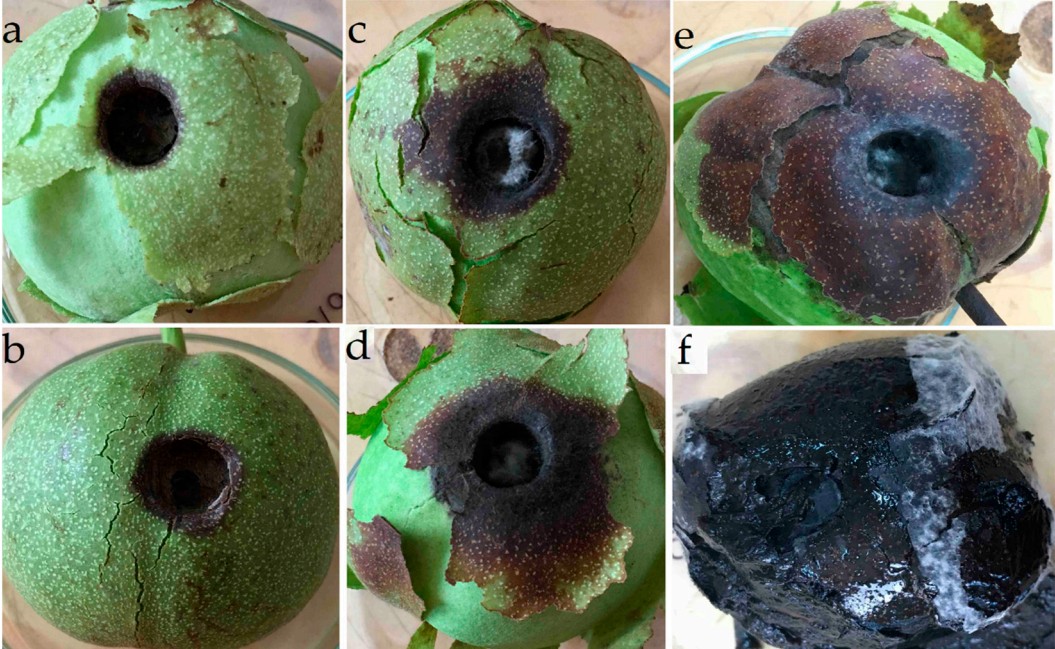

**Figure 2.** The size of the disease lesions three weeks after wounding and inoculation with fungal mycelium, and the classification of the progress of necrosis. (**a**) 0—mock wound; (**b**) 1—secondary lesion visible around the wound; (**c**) 2—more extensive brownish secondary lesion; (**d**) 3—brownish diseased tissue on half of the husk; (**e**) 4—$\frac{3}{4}$ of the fruit is affected; (**f**) 5—entirely rotten husk.

The McKinney index (Imc%), a measure of disease severity, was calculated both for the walnut shell and the kernel [53], as follows:

$$Imc\ (\%) = \frac{Sum\ of\ all\ disease\ rating}{total\ number\ of\ rating \times maximum\ disease\ grade} \times 100$$

To verify Koch's postulates [54] for assessing causation in walnut rot, the consequences of the inoculation of healthy walnut fruit with life fungal biomass were investigated with the fungal monocultures obtained, as described in Section 2.1.

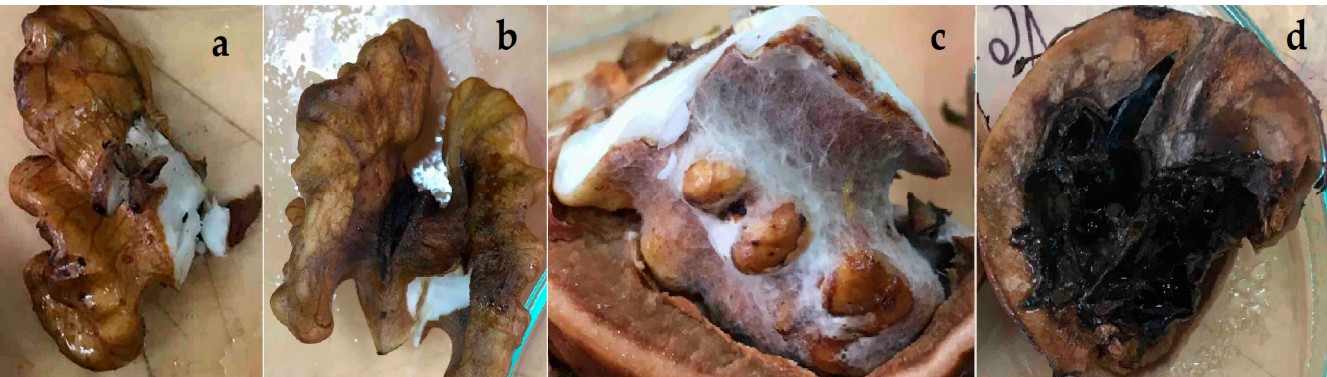

**Figure 3.** Symptoms on kernels and the classification of the progress of necrosis three weeks after inoculation onto a freshly inflicted wound of the husk. (**a**) 0—symptomless fruits; (**b**) 1—discoloured kernel; (**c**) 2—kernel covered with mycelium; (**d**) 3—entirely rotten nuts.

## 3. Results

### 3.1. Morphological and Genetic Characterization of Fungal Isolates

During the field surveys, external symptoms of canker and dieback were observed on the twigs, buds, and shoots, albeit the general condition of the trees in the four orchards was suitable to investigate the aetiology of walnut rot emerging in the region. Altogether, 56 samples were collected from 40 trees (Table 1). The majority of the samples (40) were shoots (twigs), while seven green fruits and nine buds were taken. In addition to discoloured, cankered, and pycnidia-covered plant samples, seemingly healthy plant segments were also collected (Figure 4).

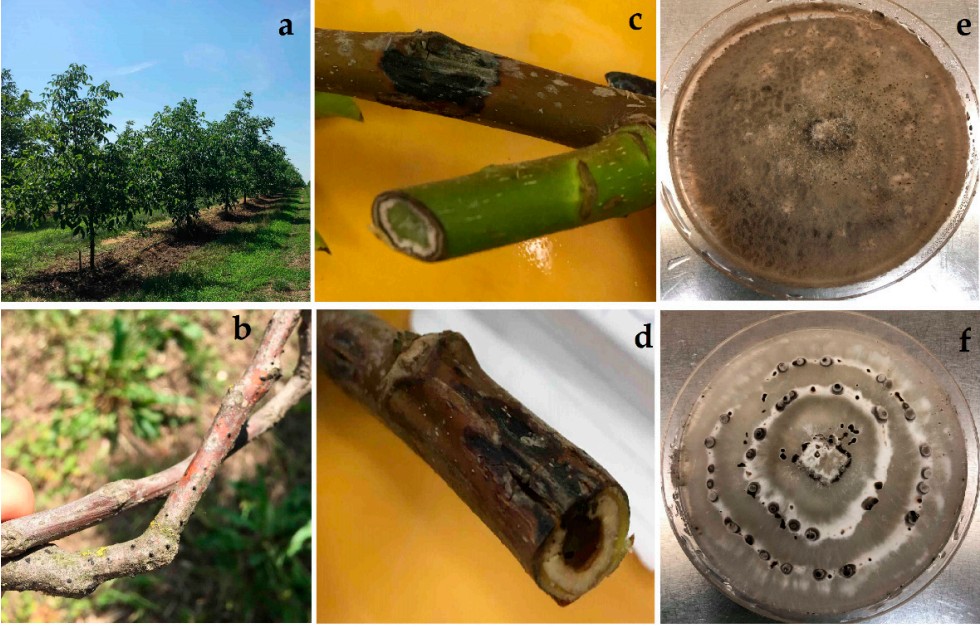

**Figure 4.** (**a**) Walnut orchard in Hajdúdorog. (**b**) Walnut twig with black pycnidia. (**c**) External lesions. (**d**) Internal lesions. (**e**) *Diplodia seriata* on the PDA medium. (**f**) *Diaporthe eres* on the PDA medium.

In the case of the symptomatic samples, internal brown lesions were visible in the twigs after sectioning (e.g., Figure 4d). Examination of the obtained 113 pure fungal monocultures revealed that *Diaporthe* sp. and a species of the family Botryosphaeriaceae (later identified as *D. seriata*) were represented by seven (6.2%) isolates each. *Diplodia* sp. and *Diaporthe* sp. were present in/on 17.5% of the sampled trees and in 25% of the 56 analysed factual samples (Table 1), predominantly in the tissues with disease symptoms. Seven fungal isolates showed phenotypic characteristics of *Diplodia* sp. [21] and seven others of *Diaporthe* sp. [27,45]. All of the isolates were originated from different trees (i.e., 35% of the sampled trees). Two *Diplodia* sp. isolates were cultured from an asymptomatic bud and a symptomatic green fruit, while other *Diplodia* and all *Diaporthe* isolates were grown from symptomatic shoots (young twigs) (Tables 1 and 2).

**Table 2.** Isolates cultured from walnut plant parts and accession numbers of the sequenced ITS and *tef1* PCR fragments.

| Species | ID of Isolates [1] | Plant Part | Geographical Origin | GenBank Accession Number [2] | |
| --- | --- | --- | --- | --- | --- |
| | | | | ITS | *tef1* |
| *Diaporthe eres* Nitschke | J1004 | Symptomatic twigs | Jánkmajtis, Hungary | MN726700 | ON381300 |
| | T1010 | Symptomatic twigs | Tarpa, Hungary | MN726702 | ON381301 |
| | U1001 | Symptomatic twigs | Újfehértó, Hungary | MN726698 | ON381302 |
| | U1003 | Symptomatic twigs | Újfehértó, Hungary | MN726699 | ON381303 |
| | U1008 | Symptomatic twigs | Újfehértó, Hungary | MN726701 | ON381304 |
| *Diplodia seriata* De Notaris | D1012 | Asymptomatic bud | Hajdúdorog, Hungary | MN726703 | ON381305 |
| | U1012 | Symptomatic twigs | Újfehértó, Hungary | MN726705 | OP207880 |
| | U1013 | Symptomatic green nut | Újfehértó, Hungary | MN726704 | OP207879 |

[1] First letters indicate the origin of the samples. J: Jánkmajtis, T: Tarpa, U: Újfehértó, D: Hajdúdorog.
[2] ITS = internal transcribed spacer, *tef1* = translation elongation factor 1-α.

When grown on a PDA medium, *Diplodia* colonies were initially white then became olivaceous green, and later turned to greenish grey with cottony texture (Figure 4e). Fast-growing mycelia were observed in abundance [21]. The growth of *Diaporthe* spp. isolates was relatively slow, the mycelia were white and became dirty white, then brown coloration appeared in the middle with solitary conidiomata visible after 5 days of incubation. After 10 days of incubation, black pycnidia were observed in concentric rings, a phenotypic character common to *Diaporthe* species (Figure 4f) [27,45].

Two *Diplodia* sp. isolates were cultured from a bud and a green fruit, while the other purified *Diplodia* and all of the *Diaporthe* isolates were grown from shoots (young twigs) (Table 2). The sequence analysis of the nuclear ribosomal internal transcribed spacer (ITS) and the large intron in the *tef1* gene (viz. the ubiquitous gene encoding translation elongation factor EF-1 *alpha*) PCR fragments (see Section 2.2 for experimental details) confirmed the presence of *Diplodia seriata* De Notaris and *Diaporthe eres* Nitschke in the selected monocultures, the results of which supported the earlier preliminary identification by morphological characteristics (Table 2).

All of the five analysed *Diaporthe* isolates were closely associated with the *D. eres* species complex in the *tef1* phylogeny, clustered with other *D. eres* Nitschke strains and with other *Diaporthe* species [55] (*D. celestris* and, to a lesser extent, *D. celastina* and *D. bicinta*) (Figure 5B; Table 2). However, a recent study by Hilário and co-workers provided evidence that this "complex of species" actually represent a single species [56]. Therefore, all of our Hungarian isolates can be considered as strains of *D. eres* Nitschke.

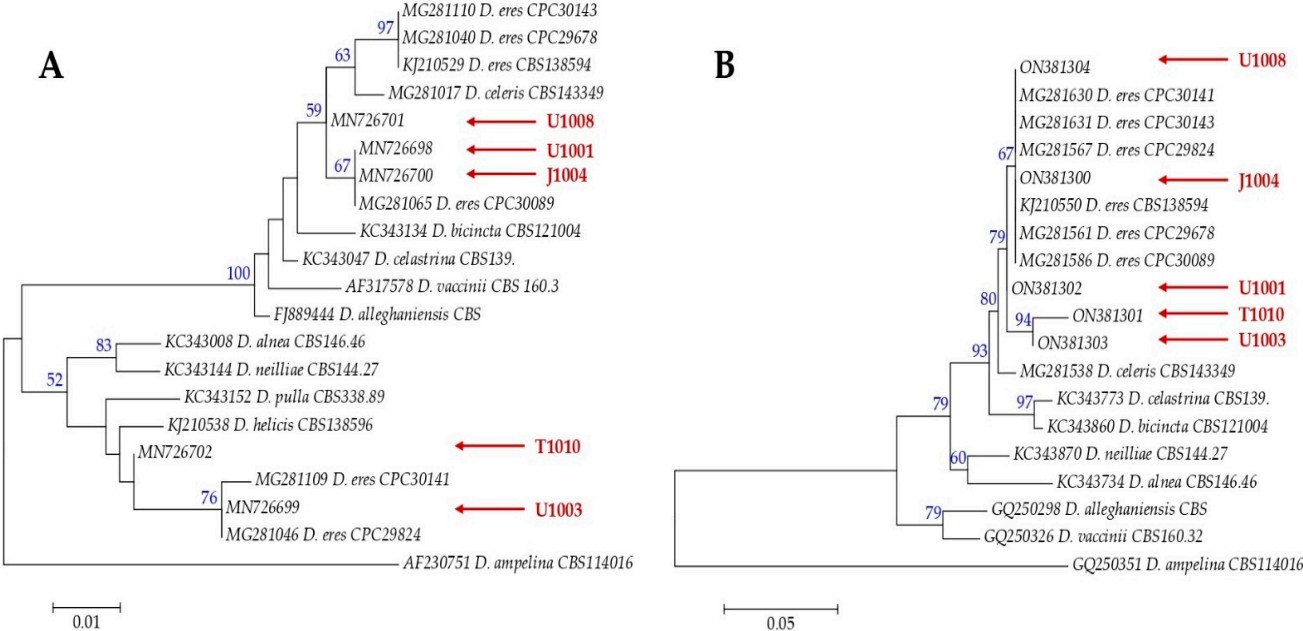

**Figure 5.** Maximum likelihood phylogenetic trees of the *Diaporthe eres* species complex. (**A**) ITS and (**B**) *tef1* sequences. Sequences deposited at the NCBI's webservers with an accession number before species name (and strain number), are from [39] and this study (Table 2). Our isolates cultured from walnut are indicated in red. Strain CBS 138594 is a culture of the epitype *Diaporthe eres*. The length of branches is proportional to the number of nucleotide differences per site, and the scale is shown under the dendrogram. Branch support values (>50%) resulting from 1000 iterative bootstrap replicates are given above the nodes. The trees are rooted on *Diaporthe ampelina* (Berk. and M.A.Curtis) R.R.Gomes, Glienke, and Crous (CBS114016).

Two of the selected *Diplodia* isolates were clustered with the *D. seriata* De Notaris epitype strain CBS 112555 in the phylogenetic analysis with concatenated ITS and *tef1* marker sequences of fungi within the *Diplodia seriata* species complex (Figure 6 and Table 2). For the ITS marker, the sequences of two of the Hungarian isolates were 100% identical to the orthologue sequence in CBS 112555 (results not shown). The other isolate clustered with another group of *D. seriata* isolates, recently defined by Zhang and co-workers [49].

### 3.2. Inoculation of Healthy, Green Walnut Fruits

The pathogenicity test protocol using monoclonal cultures of the species purified from the collected walnut tree materials is detailed in the Materials and Methods Section 2.3. After *in vitro* inoculation, four of each five green fruits tested (80%) became entirely rotten within three weeks. All of the tested *Diaporthe* and *Diplodia* isolates produced large, dark secondary lesions on the originally green husks (Figure 7).

The symptoms the on kernels appeared to be milder or developing/progressing slower than those on the nutshell and varied from the visible mycelial outgrowth and spots of necrotic tissue (Figure 7c) to complete necrosis. The results were essentially identical in each of the three biological replication experiments; the pathogens induced the same symptoms of rot and progressed at a comparable rate. The infection controls, where the deliberate wound lesions were mock inoculated with sterile PDA plugs, remained symptomless. The quantified infection rates for the severity of symptoms caused by inoculated pathogens are summarized in Table 3. We observed that *D. eres* isolates caused more severe symptoms of necrosis both on husks and kernels, or the disease progressed faster in the experimental set-up than in the case of the *D. seriata* isolates. The Sordariomycetes pathogen thus seemed to be more aggressive to walnut fruit than the Dothideomycetes pathogen, although both caused entirely rotten fruit. Both pathogens could be re-isolated from all of the infected walnuts generated in our pathogenicity test, but not from the disinfected and mock

inoculated control fruits, which revealed a direct correlation between the development of necrotic symptoms and the causal pathogens.

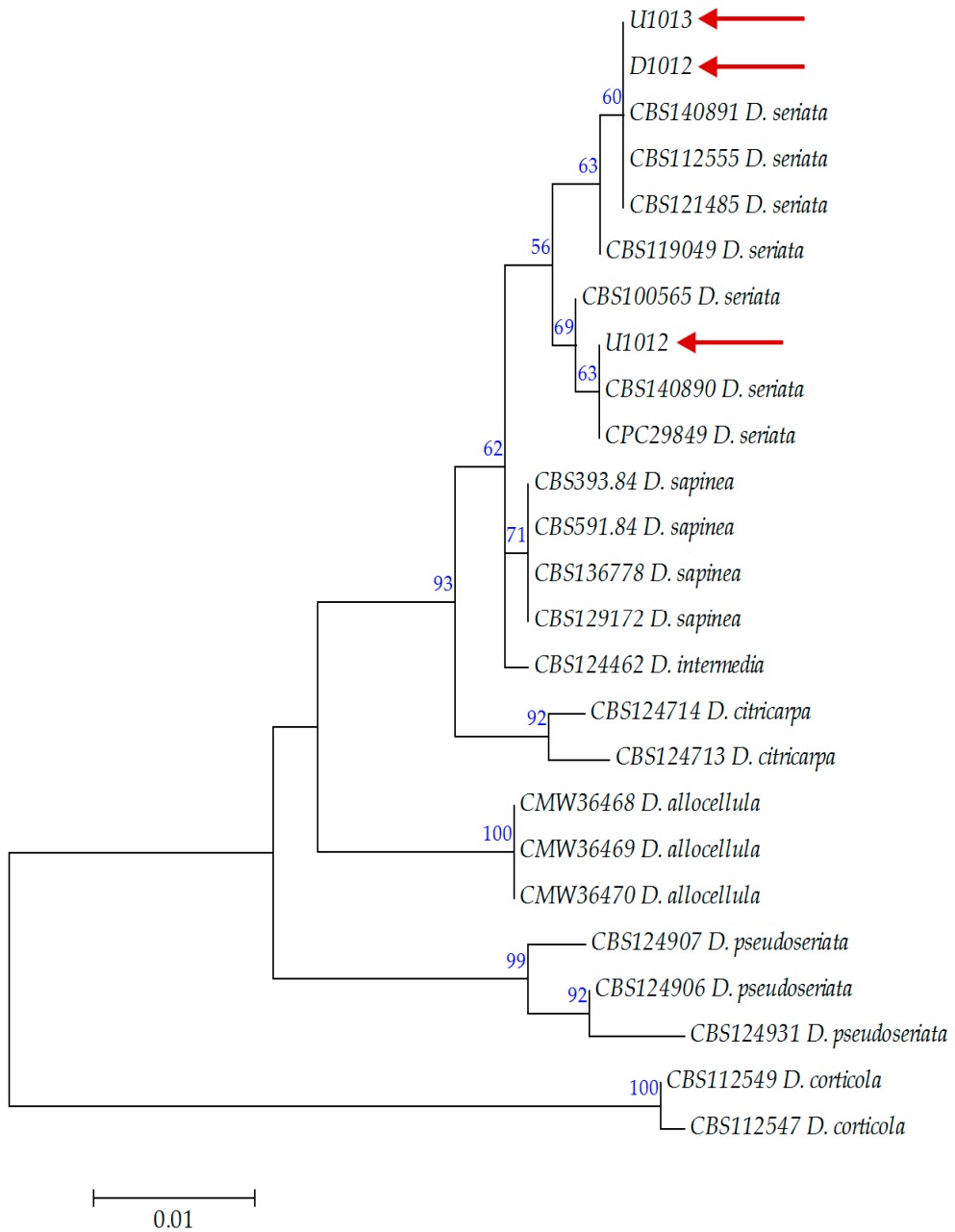

**Figure 6.** Maximum likelihood phylogeny inferred for the *Diplodia seriata* species complex for the concatenated ITS and *tef1* molecular taxonomy markers. The publicly accessible sequences of the analyzed strains from previous studies are in Supplementary Table S1 and those from this study in Table 2. The names of three monoclonal isolates cultured from walnut tree materials (this work) are indicated in red. Strain CBS 112555 is a culture of the epitype *Diplodia seriata*. Branch length is proportional to the number of nucleotide changes per site; the scale bar is shown underneath the tree. Relative bootstrap values resulting from 1000 iterative replicates are given beside the relevant nodes, where they exceed 50%. The tree is rooted on *Diplodia corticola* A.J.L. Phillips, A. Alves, and J. Luque (strains CBS 112549 and CBS 112547) orthologous sequences, respectively.

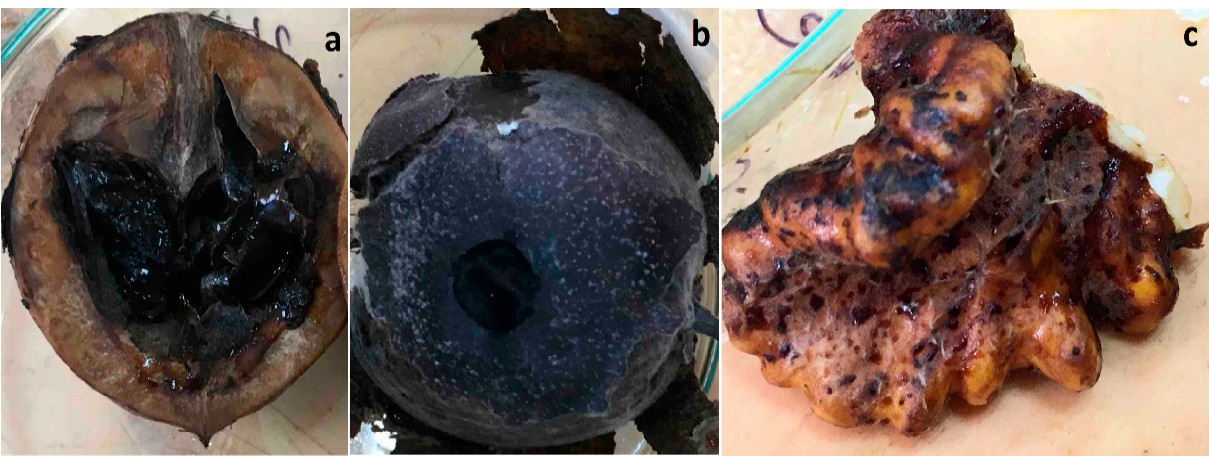

**Figure 7.** Examples of kernel (**a**,**c**) and husk (**b**) necrosis caused by *Diaporthe eres* symptomatic nut after the inoculation of the immature fruit with *Diplodia seriata*. The pictures were taken three weeks after the inoculation of a deliberately inflicted husk lesion with growing the mycelium of the pathogen taken from a PDA plate monoculture.

**Table 3.** Average McKinney index (Imc%) [53] of *Diaporthe eres* Nitschke and *Diplodia seriata* De Notaris isolates on the green husk and kernel of English walnuts. Calculated from the results recorded three weeks after the inoculation of a deliberately inflicted wound with life mycelia of the pathogens.

| | Imc% ± SD | |
| --- | --- | --- |
| | *Diaporthe eres* **Nitschke** [a] | *Diplodia seriata* **De Notaris** [b] |
| Husk | 93 ± 16 | 87 ± 12 |
| Kernel | 69 ± 16 | 50 ± 14 |

[a] Values represent the mean of five isolates ± standard deviance of the mean. [b] Values represent the mean of five isolates ± standard deviance of the mean.

## 4. Discussion

Increasing fruit losses have been reported in commercial walnut orchards in Hungary, a Middle European country with traditionally continental weather. A preliminary study indicated the presence of fungi from the Diaporthaceae and Botryosphaeriaceae families in rotten walnuts with 62% and 73.3% infection indexes, respectively [34]. Our study showed that phytopathogenic *Diaporthe* and *Diplodia* species were present in the walnut orchards (12% of the total isolated fungi) on the tree twigs, buds, and fruit, with and without visible symptoms of necrosis. More than a third (35%) of the 40 examined walnut trees were infected by either of these pathogens, and they could be isolated from small, pre-extant lesions in the vulnerable young plant tissue. The opportunistic behaviour of these fungi is thus indeed assumable, but without an explicit investigation into this direction, we would refrain from explicitly naming that. Representative isolates from different geographical locations and from different plant parts were chosen for the exact identification with ITS and *tef1* marker sequences. The phylogeny of these typical taxonomic marker sequences with the ortholog sequences from the strains of the related fungi within the same genera suggested that some of the strains we isolated from walnut trees in the north-east of Hungary were highly similar if not indistinguishable from *D. seriata* De Notaris and *D. eres* Nitschke epitypes, as recently described in literature published during our own investigation [37,43,49]. Both fungi are known to be pathogens of several woody host plants worldwide, including of English walnut [57–59]. Although both species were isolated previously from grapevine trunks in Hungary [35,36], this is the first study reporting the isolation and identification of these species from Hungarian walnut trees as causal agents of emerging walnut rot.

Both fungi could be isolated from the woody parts of nut trees. Cankers on walnut trees associated with *D. seriata* De Not. infection were detected in countries with Mediterranean-type climate, e.g., in Iran [60] and California [61], but lately walnut rot has been reported to occur in the Czech Republic [62]. In Italy, *Diplodia* teleomorph *Botryosphaeria* species caused disease in walnut orchards [43]. *Diaporthe* species are also encountered on walnut trees; the fungi of this genus were identified on *Juglans* in China [27], Spain [29] California [63], and Chile [30]. *D. eres* Nitschke causes serious necrosis in hazelnut fruits in the Caucasus and in the Czech Republic [62,64].

Developing a management strategy is difficult because of the epidemiology of *Diplodia* and *Diaporthe* genera. Pycnidia, pseudothecia, and spores are able to overwinter in the tissues of all parts of nut crop trees, including in pruning debris left on the ground [29]. The growth cracks, pruning, and other wounds can serve as an entry point for opportunistic fungi. Rainy weather exacerbates the development and dissemination of fungal spores, which can cause large-scale proliferation of phytopathogens [20,29]. As a higher temperature also benefits fungal development and proliferation, the typical necrosic symptoms on the trees often appear only in late spring [20].

We performed artificial inoculation tests on immature walnut fruit to study the pathogenicity of our new *Diaporthe* and *Diplodia* isolates. *D. eres* Nitschke and *D. seriata* De Notaris isolates from twigs and buds were selected to define their virulence in vitro on immature *Juglans regia* L. fruits. The McKinney index was higher in the case of *D. eres* Nitschke, although *D. seriata* De Notaris caused similarly serious necrosis on/in fruits. The test proved that the two pathogenic fungi, colonizing or lingering on different parts of walnut trees, were able to infect and cause disease symptoms on/in immature fruits.

*D. eres* Nitschke and *D. seriata* De Notaris were isolated from the shoots, buds, and walnut fruit, indicating a complex disease aetiology and the potential for nearby sources of the pathogens. In line with our results, these pathogens may be able to penetrate the trees through surface injuries all year around and eventually reach the seasonal walnut fruits, and cause them to rot while still attached to the tree. This latter process is reportedly correlated with the prevalent weather conditions [20,29]. Furthermore, this mode of step-wise infection can work in a reverse direction; infected fruits can transmit the pathogens in the form of condiospores to other plant parts as well as to neighbouring, healthy trees [61]. The potential for such cross-infection events is known to be from regions with a Mediterranean climate, such as California, Italy, and southern Spain [20,29], but has never been investigated in regions with continental weather, such as the Carpathian Basin.

Our results highlight that appropriate horticultural practices (e.g., sanitation) may restrict or control walnut fruit rot outbreak and tree canker. To define adequate management practices, it is crucial to understand the aetiology of this type of diseases. Further research is required, for instance, to determine the optimal growth temperature of pathogens, in order to study the synergistic interactions, as well as a comprehensive investigation of the tree microbiota in different seasons. The timing and method of tree pruning could be key to limit the spread of disease, as the resulting wounds are appropriate infection sites for pathogens. The application of fungicides is now common practice after pruning in some nut orchards. Not only purposeful wounds resulting from branch pruning provide occasions for infection en masse, but damage caused by high winds, hail, heat waves, drought, frost, or by pests [20,29] may equally present easy infection opportunities. The spread of the walnut husk fly (*Rhagoletis completa* Cresson) [65] may contribute to increasing walnut fruit loss in the near future.

## 5. Conclusions

Diaporthaceae and Botryosphaeriaceae species are known pathogens of woody plants with a worldwide occurrence [20,22,23,25,28,33,34,66]. These phytopathogens are often reported as causative agents of twig canker, dieback, and shoot blight on English walnut (*Juglans regia* L.), and may be responsible for nut rot as well [19,21,24,26,27,29,30,42]. *D. eres* Nitschke and *D. seriata* De Notaris were isolated from buds and twigs from Hungarian

walnut orchards in four different localities in the north-east of the country and caused serious necrosis symptoms in immature green nut, following artificial, in vitro inoculation in deliberately inflicted wound lesions. The results provide indications that fungal conidia or mycelia locally occurring in the same tree or a tree in its direct neighbourhood, surviving all year round on (permanent) woody plant parts, can cause infection in surface wounds on developing nuts in the production season [21,29,30], not only in countries with Mediterranean weather [20,29,43,63], but also in walnut producing regions with traditionally continental weather, such as Hungary. Those pathogens [20,21,30] can cause severe damage to stressed plants [25,37]. One consequence of climate change and the associated changing weather patterns is an increase in infection rates, exacerbating economic damage caused by these pathogens in all walnut producing regions.

**Supplementary Materials:** The following supporting information can be downloaded at: https://www.mdpi.com/article/10.3390/horticulturae9020205/s1. Table S1: Isolates and sequences from other studies used in phylogenetic analyses for this study.

**Author Contributions:** Conceptualization, A.Z., F.T. and E.S.; data curation, K.P., E.F. and M.F.; formal analysis, C.K., K.P., L.K. and E.S.; investigation, C.K., K.P. and E.F.; methodology, A.Z., K.P., E.F., K.M., M.F. and E.S.; resources, A.Z., F.T., L.K. and E.S.; supervision, E.S.; visualization, A.Z., K.M. and M.F.; writing—original draft, A.Z., C.K., F.T., K.P., F.P., E.F., L.K., K.M., M.F. and E.S.; writing—review and editing, F.P., L.K., M.F. and E.S. All authors have read and agreed to the published version of the manuscript.

**Funding:** This research was supported by the National Research, Development, and Innovation Fund (grants: NN 128867 to L.K. and K 138489 to E.F.).

**Data Availability Statement:** The datasets used in the current study are available from the corresponding author upon reasonable request.

**Acknowledgments:** The authors are grateful to Beáta Tóth and Gyula Szakadát for their technical support.

**Conflicts of Interest:** The authors declare no conflict of interest.

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
