# Peer review of "Diaporthe and Diplodia Species Associated with Walnut (Juglans regia L.) in Hungarian Orchards"

_horticulturae, doi:10.3390/horticulturae9020205_

Round 1
Reviewer 1 Report (New Reviewer)
This manuscript firstly described both Diaporthe eres and Diplodia seriata fungi causing disease in walnut (according to USDA database). However, there are some points need to be addressed:
1. Title should specify which disease of walnut
Diaporthe and Diplodia species associated with...(disease).. in walnut..
2. Family name should not be italic, please correct throughout this manuscript
3. For morphological identification, have you measured dimension of spore/conidia? because morphological characteristic is an important step to identify fungal species.
4. Have you re-isolated the fungi from inoculated walnut? and the result showed the similar morphology with Diaporthe and Diplodia?
5. Discuss why D. eres cause more severe symptom than D. seriata too.
6. Omit to use author name after scientific name
7. What is condiospore?
8. Conclusion part is weak, please re-write and minimize, ovoid to repeat result and discussion in conclusion part
9. Academic English used in this manuscript need to proof.
Some minor points can be found in pdf. file

Author Response
Please see the attachment.

Reviewer 2 Report (New Reviewer)
Dear authors,
I have carefully revised your work, identifying Diaporthaceae and Botryosphaeriaceae species in plant parts other than fruits in walnut trees and evaluating their pathogenic capacity on healthy fruits.
I consider the manuscript excellent for publication since a very careful use of English was taken and the scientific information was transmitted in a very comprehensive and detailed manner.
Author Response
Your opinion is appreciated, thank you very much indeed!
Reviewer 3 Report (New Reviewer)
Introduction is very well written and informative. It shows a clear review of the importance of walnut trees and nuts and of the diseases affecting their health with a lot of relevant citations. At the end of introduction, the aims of the study are clearly stated. Only minor changes are needed in this section, mostly regarding the English language. Please see the comments in the pdf file for the details.
Materials and methods
Some steps of the sample collection and analysis should be more thoroughly explained. Please see the comments in the pdf file for details. Besides that, also very well written and excellent photos of symptoms on inoculated walnut fruits.
Results
This section requires minor changes as well, mostly regarding the details about the link between the obtained fungal isolates and the samples they came from. Please see the comments in the pdf file for details.
Discussion and Conclusions
There is an entire chapter repeating twice, this should be resolved. Also, there were 14 fungal isolates belonging to Diaporthe or Diplodia found in analysed samples, however, only eight of them were subjected to molecular analysis and pathogenicity testing. This should be explained in discussion and in the materials and methods section as well. Also, the fact that Diaporthe/Diplodia are opportunistic fungi, often living in the healthy tissues of plants, should be more emphasized in these sections. Other than that, this paper brings new and interesting results and conclusions and can serve as a template for new research and guidelines for the walnut orchards management.

Author Response
Please see the attachment.

This manuscript is a resubmission of an earlier submission. The following is a list of the peer review reports and author responses from that submission.
Round 1
Reviewer 1 Report
Manuscript title: The etiology of walnut (Juglans regia L.) pathogen fungi in Hungarian orchards
Manuscript id: horticulturae-1734213
Authors: Zabiák et al.
The manuscript is great regarding less studied topic and the experimental set up on diseases of walnut tree. The manuscript regarding the topic and results presented is of interest to plant scientific community and revisions based on the comments below are recommended before considering for publication.
Major comments
- Insufficient abstract: In the abstract, the main aim of the manuscript is missing, the current version it only highlights the result. In addition, it would be even better to have a sentence as future perspective.
- The DOI of some of the references are missing…..
- Line 60-67, the aim or hypothesis of the study is mixed or repeated clear, and the approach is missing ….
- Lake of scientific literature to support the statements and finings throughout the manuscript…... I have made some suggestions for that and more need it….
- More information needed for ALL TABLE captions and define the abbreviation and units that used. And adjust the significant figures for the table and manuscript.
- Grammar and punctuation issuers are need to be addressed. I have selected/mentioned some as example.
- I am not sure whether the ‘etiology’ term is well discussed in the abstract and manuscript. Please consider discuss it or rephase it.
- I have a major concern about the results and discussion section. The authors describe results and compare the results with previous studies, however, insight mechanisms are still not sufficient.
- In introduction, a general classification or discussion of the diseases are missing, please consider grouping the diseases or briefly discuss them.
Minor comments:
Abstract:
Line 15-17: The statement is not discussed in the manuscript, why should we have it in the abstract, for e.g. weather conditions.
Introduction:
Line 37-38: consider using this reference: https://doi.org/10.1080/22297928.2016.1152912
Line 40-43: A complicated sentence, please revise and check grammar
Line 40: You have used ‘’weather factors’’, however in line 16 you used ‘’ weather conditions’’ Please consider harmonize the term throughout the manuscript. Two terms in the same manuscript are confusing.
Line 42-47: A reference needed here.
Line 41-43: Two conjunctions (too, and) with the same meaning in one sentence. Please rephrase.
Line 57-59: Consider using the following references:
https://doi.org/10.1094/PDIS-03-19-0545-RE
https://doi.org/10.1094/PDIS-07-13-0706-RE
In MM section
Literature references are missing for all sub-section. It would be better to cite the references that the procedure adapted.
Line 116-118: What are the controls? How did you prepare them? A detail description needed here as its very crucial for the experimental part.
122-128: A demonstration of the color is needed, consider adding a figure in the manuscript or in the supplementary file to show the color scale, the text version is not visualize the scaling.
R&D section
172-178: The correlation between the isolates and sampling location is missing? Consider to add the correlation, it’s OK if there is no correlation, then we know the impact of location is none.
Line 189-190: How the controls are assessed? And analyzed? Please elaborate more on the assessment of the controls.
Table 3: is the data from all locations?? Is not please add specific location.
Line 227-247: This section is repeating information already presented and explaining things in an unnecessarily complicate way. The quality of the manuscript would benefit from the whole section being condensed.
Line 222-226 and line 235-244 belong to result section, if you decided to keep the result and discussion sections separate, then move them to result section.
Line 246: ‘’tree cancer’’ is a new term, consider use the correct term.
Line 249-255: A complicated sentence, please revise and check grammar
Line 255-257: A reference needed here.
Line 258-262: A complicated sentence, please revise and check grammar
Line 264-267: A reference needed here.
The conclusion section is missing, I believe there are other a lot nice conclusions could be made from this study…. And the future perspectives for following research highly crucial here....
Author Response
Major comments
- Insufficient abstract: In the abstract, the main aim of the manuscript is missing, the current version it only highlights the result. In addition, it would be even better to have a sentence as future perspective.
Response:
Abstract were corrected, indicating the aim and about the future perspectives.
- The DOI of some of the references are missing…..
Response:
References were carefully checked, however the following publications has no DOI:
Boriss, H.; Brunke, H.; Kreith, M. Commodity Profile: English Walnuts. Agricultural Issues Center 2006, University of California. Davis. (https://aic.ucdavis.edu/profiles/Walnut-2006.pdf) (accessed on 29 april 2022)
FAOSTAT Crops and livestock products. http://www.fao.org/faostat/en/#data/TP (accessed on 15 April 2022)
McKinney, H. H. Influence of soil temperature and moisture on infection of wheat seedlings by Helmintosporium sativum. Journal of Agricultural Research 1923, 26, 195–218.
Kovács, Cs.; Peles, F.; Bihari, Z.; Sándor, E. Endophytic fungi associated with Grapevine Trunk Diseases, from Tokaj wine region, Hungary. Növényvédelem 2014, 50, 153-159.
Michailides, T. J.; Chen, S. F.; Morgan, D.; Felts, D.; Nouri, M. T.; Puckett, R.; Luna, M.; Hasey, J.; Anderson, K.; Coates, W.; Fichtner, E.; Buchner, R.; Bentley, W. Managing Botryosphaeria/Phomopsis cankers and anthracnose blight of walnut in California. In Walnut Research Reports. California Walnut Board: Folsom, CA, 2013,pp. 325-346. http://walnutresearch.ucdavis.edu/2013/2013_325.pdf (accessed on 29 april 2022)
Tóth, M.; Nagy, A. Szanyi, Sz.; Kiss, O. M.; Voigt, E. Field study of the synthetic pheromone lure of the walnut husk fly (Rhagoletis completa cresson) (Diptera: Tephritidae) on three Rhagoletis spp. Növényvédelem 2021, 82 [n. S. 57]: 5. 201-207.
ISBN/ISSN numbers were provided for the following references:
Holb I. A dió kórokozói. In A héjasok növényvédelme, Editor Radócz, L. Szaktudás Kiadó Ház: Budapest, Hungary, 2002, pp. 28-48. ISBN: 9789638617088
Barr, M. E. Prodromus to Class Loculoascomycetes. Hamilton I. Newell, Inc., Amherst 1987, 82-86. ISBN: 0-934454-51-5
Michailides, T. J.; Morgan, D. P. Association of Botryosphaeria panicle and shoot blight of pistachio with injuries of fruit caused by Hemiptera insects and birds. Plant Disease 2016, 100, 1405–1413. DOI:10.1094/PDIS-09-15-1077-RE
White, T. J.; Bruns, T.; Lee, S.; Taylor, J. W. PCR Protocols: A Guide to Methods and Applications. Academic Press, New York, USA, 1990, pp. 482. ISBN: 978-0123721815
Phillips, A. J. L.; Crous, P. W.; Alves, A. Diplodia seriata, the anamorph of “Botryosphaeria” obtusa. Fungal Diversity 2007, 25, 141-155. ISSN 1560-2745
- Line 60-67, the aim or hypothesis of the study is mixed or repeated clear, and the approach is missing ….
Response:
The aim of the study was modified:
The aim of this study were to (i) detect the presence of these fungi on different plant parts (twigs, buds, shoots) in commercial walnut orchards in Hungary; to (ii) identify them using molecular marker sequences ITS and tef1; and (iii) to test their pathogenicity on green walnut.
- Lake of scientific literature to support the statements and finings throughout the manuscript…... I have made some suggestions for that and more need it….
Response: Additional scientific literature were added to the manuscript.
- More information needed for ALL TABLE captions and define the abbreviation and units that used. And adjust the significant figures for the table and manuscript.
Response: Information were added for table captions. Figures were adjusted.
- Grammar and punctuation issuers are need to be addressed. I have selected/mentioned some as example.
Response: Punctuation was carefully checked.
- I am not sure whether the ‘etiology’ term is well discussed in the abstract and manuscript. Please consider discuss it or rephase it.
Response: The ‘etiology’ term is discussed in the introduction.
“In recent years nut crops are significantly affected by fungal diseases, e.g. twig canker, dieback, shoot blight and nut rot [17,18,19,20,21]. The aetiology studies proved, that species of the Botryosphaeriaceae and Diaporthe family were often reported as causative agents of those symptoms on woody plants, like walnut in several countries [19,20,21,22,23,24,25,26,27,28]. These pathogens are able to survive on inactive and dead plant parts and produce conidia in pycnidia. Spores than infect wounds on different plant parts, including nuts [19,27,28].”
- I have a major concern about the results and discussion section. The authors describe results and compare the results with previous studies, however, insight mechanisms are still not sufficient.
Response: Discussion part was rewrite following reviewers’ suggestion.
- In introduction, a general classification or discussion of the diseases are missing, please consider grouping the diseases or briefly discuss them.
Response: Diseases were classified and briefly discussed.
Minor comments:
Abstract:
Line 15-17: The statement is not discussed in the manuscript, why should we have it in the abstract, for e.g. weather conditions.
Response: Abstract was modified, the mentioned sentence was discarded. Weather conditions are mentioned in the following sentence:
"Climate change, resulting extreme weather conditions and the reduction in the use of pesticides in the European Union forecasts growing number of infections and increasing economic damage caused by these pathogens."
Introduction:
Line 37-38: consider using this reference: https://doi.org/10.1080/22297928.2016.1152912
Response: Reference was added
Line 40-43: A complicated sentence, please revise and check grammar
Response: It was revised
Spring frosts can cause some problems during the fertilization due to cold sensitivity of the generative organs. The damage results in paper-nutshell with hollows or even extinction of the kernel [10,11].
Line 40: You have used ‘’weather factors’’, however in line 16 you used ‘’ weather conditions’’ Please consider harmonize the term throughout the manuscript. Two terms in the same manuscript are confusing.
Response: ‘’weather factors’’ has eliminated and ‘’ weather conditions’’ phrase use only in the manuscript.
Line 42-47: A reference needed here.
Response: References were added.
Line 41-43: Two conjunctions (too, and) with the same meaning in one sentence. Please rephrase.
Response: Sentence was rephrased: Nut production can be influenced by weather factors, insects and microbes.
Line 57-59: Consider using the following references:
https://doi.org/10.1094/PDIS-03-19-0545-RE
https://doi.org/10.1094/PDIS-07-13-0706-RE
Response: References were added.
In MM section
Literature references are missing for all sub-section. It would be better to cite the references that the procedure adapted.
Response: References were added to all methods.
Line 116-118: What are the controls? How did you prepare them? A detail description needed here as its very crucial for the experimental part.
Response: Description about preparation of the control was added:
"Fruits were inoculated through a wound with mycelial plugs from the margin of seven days old fungal cultures [27]. The control consisted cork borer wounded husk inoculated with sterile PDA plug, in three replicates. They were incubate among the same conditions as pathogens inoculated samples, but separately from them, in a different plastic box. The inoculated green walnuts were then sealed with Parafilm (Merck, Germany) in order to avoid dehydration and incubated at 25 °C for three weeks [19]. "
122-128: A demonstration of the color is needed, consider adding a figure in the manuscript or in the supplementary file to show the color scale, the text version is not visualize the scaling.
Response: Figures were added to demonstrate the scaling categories in case of hull and for kernel.
The conclusion section is missing, I believe there are other a lot nice conclusions could be made from this study…. And the future perspectives for following research highly crucial here....
Response: Conclusion was added.
Reviewer 2 Report
The manuscript “The etiology of walnut (Juglans regia L.) pathogen fungi in Hungarian orchards” presents relevant information and can be considered for publication in Horticulturae journal with minor revisions.
The authors should write the species name in italics (e.g., Juglans regia L.; Diplodia seriata De Not.). The authors should write out in full the genus and species, both in the title of the manuscript and at the first mention of an organism in a paper. After first mention, the first letter of the genus name followed by the full species name may be used (e.g., J. regia L.; D. seriata De Not.).
It is advisable to consult specific databases (e.g. index fungorum) to check the current name with the relative classifiers.
Author Response
The authors should write the species name in italics (e.g., Juglans regia L.; Diplodia seriata De Not.).
Reply: Species name were put in Italics, and authority was added.
The authors should write out in full the genus and species, both in the title of the manuscript and at the first mention of an organism in a paper.
Reply: Corrected following the instruction.
After first mention, the first letter of the genus name followed by the full species name may be used (e.g., J. regia L.; D. seriata De Not.).
Answer: Corrected following the instruction.
It is advisable to consult specific databases (e.g. index fungorum) to check the current name with the relative classifiers.
Answer: Corrected following the instruction.
Reviewer 3 Report
The manuscript entitles “The etiology of walnut (Juglans regia L.) pathogen fungi in Hungarian 3 orchards” has been written good and have some comments below:
In abstract, authors should report their ITS results. Authors no given any outcome inside the abstract, why? I wonder.
In introduction part, Authors need to give information why you did this work and what is the need for molecular work as you have done inside your study as authors didn’t give any information regarding ITS, whether this work done or not previously, if not then they can suggest that in other fruits these type of work have been done and that is very much useful.
In Methodology part, Authors should write how many replicates they used for this study? In ITS also is this data in replicate of not? If yes then, how many biological replicates you have taken for this study? I am not understand that why authors not did any statistical analysis. How did the authors find results are significantly proved?
In results, Authors need to give dendrogram where it is showing that tef1 gene analysis confirmed the presence of Diplodia seriata and Diaporthe eres.
I am wonder to see the manuscript that authors not given any conclusion in the end. Please write conclusion of this study.
References should be according to the journals guideline. And follow same format for all the references.
English language should be improved in throughout the manuscript.
Author Response
In abstract, authors should report their ITS results. Authors no given any outcome inside the abstract, why I wonder.
Response: Seqence analysis was added to the abstract
„ Diaporthe eres Nitschke and Diplodia seriata De Not. were identified from different symptomatic and asymptomatic plant parts of walnut trees based on ITS and tef1 sequences..”
ITS sequences have been deponated in GeneBank, as indicate in Table 2. and in the following sentence in section 2.2:
“The identified sequences were deposited in the GenBank under the following ID numbers: MN726698 – 726705 and ON381300 – 381305.”
In introduction part, Authors need to give information why you did this work and what is the need for molecular work as you have done inside your study as authors didn’t give any information regarding ITS, whether this work done or not previously, if not then they can suggest that in other fruits these type of work have been done and that is very much useful.
Response: The requested information about information why this work was done, and what was the need for molecular work and previous results on grapevine was added to the introduction:
“Morphological characters are variable and not informative for species level identification of Diaporthe [29] and Botryosphaeriaceae species [30]. Molecular identification with sequence analysis of the internal transcribed spacer (ITS) of the rDNA combined with the large intron of tef1 gene encoding translation elongation factor 1 (tef1) sequences for both taxonomic groups [27,29,30,31,32,33,34].
Walnut production has high economic importance in Hungary, as its’ production area is the third largest after apple and sour cherries [35], with continuously increasing production [36]. Diaporthe and Botryosphaeriaceae species have become widely distributed in Hungarian commercial walnut orchards an nurseries [37,38], and in the Hungarian vineyards as grapevine trunk diseases pathogens [39,40]. The importance of these pathogens may further increase due to the climate change [34,41], which suggest that Diaporthe and Botryosphaeriaceae species are a real threat to the English walnut commerce in Hungary [42].”
In Methodology part, Authors should write how many replicates they used for this study. ITS also is this data in replicate of not If yes then, how many biological replicates you have taken for this study I am not understand that why authors not did any statistical analysis. How did the authors find results are significantly proved.
Response: As described in section 2.1, fungal isolates were isolated from different plant parts and pure cultures were investigated by microscope, so morphological identification on genus/species level was accomplished. It was followed by DNA extraction and sequencing of ITS fragments in order to verify the results of the previous investigation. The PCR was performed twice on the DNA of each isolates, but only products from the second reaction were submitted to sequencing, because no size difference was found between the fragments of the first and second PCR reaction. Quality of the sequences was good, the BLAST search gave unambiguous hits. Since the results of the microscopic analysis and the sequencing were congruent, we did not find it necessary to repeat the sequencing.
Replicate number for the pathogenicity test was indicated. Calculated average SD was indicated in Table 3.
In results, Authors need to give dendrogram where it is showing that tef1 gene analysis confirmed the presence of Diplodia seriata and Diaporthe eres.
Response: All sequences of our isolates showed 100% sequence identity with the ex-type strain of a known species, therefore the isolates were considered to belong to this species. It was indicated in the manuscript.
I am wonder to see the manuscript that authors not given any conclusion in the end. Please write conclusion of this study.
Response: Conclusion was added.
References should be according to the journals guideline. And follow same format for all the references.
Response: References were corrected according to the journals guideline
English language should be improved in throughout the manuscript.
Response: English was carefully checked to improve.
Round 2
Reviewer 1 Report
I am happy to see the manuscript improved nicely. Author addressed all my comments adequately.
However to make sure the statements are supported with literature I will recommend to add citation in the following lines:
Line 36-38:
https://doi.org/10.1007/s12161-021-02203-0
Line 50-54:
https://doi.org/10.1094/PDIS.2002.86.6.599
or
https://doi.org/10.1094/PDIS-08-14-0801-PDN
Line 58:Reference 21 was used 2 times. Please re-edit this!
Overall, the quality of the written text with respect to English phrasing and grammar is good and acceptable.
Best wishes,
Author Response
However to make sure the statements are supported with literature I will recommend to add citation in the following lines:
Line 36-38: https://doi.org/10.1007/s12161-021-02203-0
Response: Citation was added.
Line 50-54: https://doi.org/10.1094/PDIS.2002.86.6.599
Response: Citation was added.
or
https://doi.org/10.1094/PDIS-08-14-0801-PDN
Response: Cytation was not aded, as the article is not about apical, but root rot.
Line 58:Reference 21 was used 2 times. Please re-edit this!
Response: Line 58 was edited.
Reviewer 3 Report
Dear Authors
Now Paper is looking very nice and more scientific information have been added after the suggestions. Manuscript is now looking in good shape so in my opinion it may be published.
Author Response
Dear Reviewer,
Thank you for your previous comments and suggestions to improve the manuscript!